# Sarcomeric versus Non-Sarcomeric HCM

**Felice Borrelli [1], Maria Angela Losi [1],\*, Grazia Canciello [1], Gaetano Todde [1], Errico Federico Perillo [1], Leopoldo Ordine [1], Giulia Frisso [2] , Giovanni Esposito [1] and Raffaella Lombardi [1]**

1    Department of Advanced Biomedical Sciences, Federico II University of Naples, 80131 Naples, Italy
2    Department of Molecular Medicine and Medical Biotechnology, Federico II University of Naples, 80131 Naples, Italy
\*    Correspondence: losi@unina.it; Tel.: +39-081-726-22-21

**Abstract:** Hypertrophic cardiomyopathy (HCM) is the most common heritable cardiovascular disorder and is characterized by left ventricular hypertrophy (LVH), which is unexplained by abnormal loading conditions. HCM is inherited as an autosomal dominant trait and, in about 40% of patients, the causal mutation is identified in genes encoding sarcomere proteins. According to the results of genetic screening, HCM patients are currently categorized in two main sub-populations: sarcomeric-positive (Sarc+) patients, in whom the causal mutation is identified in a sarcomeric gene; and sarcomeric-negative (Sarc−) patients, in whom a causal mutation has not been identified. In rare cases, Sarc− HCM cases may be caused by pathogenic variants in non-sarcomeric genes. The aim of this review is to describe the differences in the phenotypic expression and clinical outcomes of Sarc+ and Sarc− HCM and to briefly discuss the current knowledge about HCM caused by rare non-sarcomeric mutations.

**Keywords:** hypertrophic cardiomyopathy; sarcomeres; genetic screening; phenotypic expression; outcome

## 1. Introduction

Hypertrophic cardiomyopathy (HCM) is the most common genetic heart disease, with a prevalence of 1 in 500 individuals; it is characterized by asymmetric left ventricular (LV) hypertrophy, which is not explained by an overload caused by other cardiac or systemic diseases [1–3]. HCM has been the focus of intense clinical and basic science investigation for decades, since the discovery of the first mutation in the Myosin Heavy Chain-7 (*MYH7*) gene, encoding β-cardiac myosin heavy chain (β-MHC), in 1989 [4]. Research studies conducted over the last 30 years have provided significant insights into the genetic and pathogenic basis and clinical course of HCM. Mutations in genes encoding sarcomere proteins have been identified as the main cause of HCM and the molecular mechanisms of the disease have been largely elucidated, with the consequent improvement of clinical management and the discovery of novel therapies [5]. However, despite the progress that has been made in genetic screening, the causal mutation remains unidentified in up to 60% of HCM patients [1,6].

The pathophysiologic characteristics of HCM are unexplained LV hypertrophy (LVH), left ventricular diastolic dysfunction, a high risk of arrhythmias and sudden cardiac death (SCD), and dynamic left ventricular outflow tract obstruction (LVOTO); the latter may occur in more than 50% of the patients [7]. HCM patients often present with highly variable phenotypic expressions in terms of the degree and distribution of LVH and its clinical course. It is common to observe patients who share the same gene mutation and belong to the same family to develop a variable phenotype ranging from almost asymptomatic forma to severe arrhythmias or the evolution of heart failure. The phenotypic variability can be related to genetic, epigenetic, and environmental factors. It has been shown that the HCM phenotype may be influenced by modifier gene(s), altering the effect of causal genes [8]. Furthermore, polymorphic variants in several genes may regulate wall thickness

and prognosis in HCM, as shown by recent genome-wide association studies that have identified significant susceptibility loci for HCM [9] and suggest that common variants may explain some of the variable expressivity of the pathogenic sarcomere variants [10,11].

A diverse array of symptoms may occur in HCM patients, ranging from exertional dyspnea, fatigue, palpitations, lightheadedness, syncope, and atypical chest pain, to SCD [12,13]. Although a significant proportion of HCM patients remain asymptomatic or minimally symptomatic throughout life, SCD may occur in about 2% of patients as the first symptom of the disease; moreover, in about 8% of HCM patients, the disease may evolve towards heart failure with wall thinning, cavity enlargement, and systolic dysfunction [13]. Remarkably, signs and symptoms do not necessarily correlate with the degree of LVH or with the severity of LVOTO [14]. The poor correlation between the severity of symptoms and physio-pathological abnormalities and phenotypic variability might impair accurate and early diagnoses and be responsible for inappropriate therapy choices.

Determining the relationship between the genotype, phenotype, and outcomes over a lifetime is critical to improving risk stratification and to guiding patient clinical management.

## 2. Definitions of Sarcomeric and Non-Sarcomeric HCM

According to the results of genetic screening, HCM patients are currently categorized in two broad, relatively distinct populations: (1) the sarcomere-mutation-positive (Sarc+) group, including carriers of a sarcomere genetic mutation, which comprises about 40% of HCM patients, and (2) the sarcomere-mutation-negative (Sarc−) population, including HCM patients in whom the causal mutation has not been identified [15] (Figure 1A). In rare cases, a pathogenic variant in a non-sarcomeric gene may be identified in patients with a typical HCM phenotype: the prognosis and the clinical management of these rare cases may be assisted by the observation of the clinical course in family members and published case reports. The establishment of international registries is needed to better define the outcome and the SCD risk in this subpopulation of Sarc− HCM.

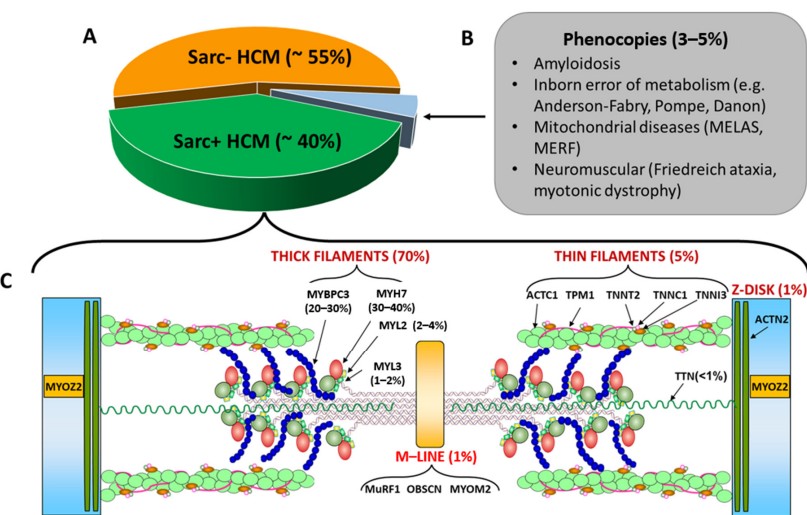

**Figure 1.** (**A**): Percentages of Sarc+ and Sarc− HCM and of phenocopies; the latter are listed in Panel (**B**). (**C**): sarcomeric genes associated with Sarc+ HCM with the relative percentages of affected individuals. The image also shows the coded pro-teins and their localization in the sarcomere. Abbreviation as in the text.

## 3. Differential Diagnosis with Phenocopies

Phenocopies (conditions mimicking the HCM phenotype, accounting for approximately 3–5% of unexplained LVH; Figure 1B) must be excluded from the Sarc− HCM sub-population [16]. These conditions include lysosomal storage disorders (e.g., Fabry disease), cardiac amyloidosis, glycogen storage disorders (e.g., Danon disease), protein kinase adenosine monophosphate-activated non-catalytic subunit gamma 2 (PRKAG) car-

diomyopathy, RASopathies (including Noonan syndrome, LEOPARD syndrome, Costello syndrome, and cardiofaciocutaneous syndrome), mitochondrial diseases, and several inborn metabolic disorders [17].

Despite being relatively less common, it is important to differentiate HCM phenocopies from both Sarc+ and Sarc− HCM because their management and prognosis differ significantly [18]. A deep understanding of the specific features of all the diseases that may cause an HCM phenotype is essential to recognizing a specific etiology [19]. Reaching a fast and definite diagnosis is crucial for the correct risk stratification of the proband and the early initiation of disease-modifying therapy (when available); moreover, the identification of the genetic causes of the disease allows for the screening of family members and the identification of additional carriers who may benefit from specific treatments [20].

The inheritance pattern and age at presentation might provide guidance for the differential diagnosis of sine causa LVH cases [19]. While rare Sarc− HCM may present as X-linked or autosomal recessive diseases, HCM typically exhibits an autosomal dominant transmission. Hence, an X-linked transmission might rather suggest a diagnosis of Anderson–Fabry or Danon disease, especially if additional signs and symptoms typical of these diseases are present, namely, acroparesthesia, gastrointestinal symptoms [21], kidney dysfunction, angiokeratomas, and anhidrosis in Fabry disease [22]; and early age of onset, muscle weakness, intellectual disability, and cardiac conduction abnormalities in Danon disease [23]. Since Sarc+ HCM is not usually associated with systemic manifestations, their presence should arouse suspicious of a different etiology. Other examples of systemic involvement are muscle weakness in mitochondrial diseases, peripheral nervous system involvement, and carpal tunnel syndrome in amyloidosis, gait disturbances in Friedreich's ataxia, facial dysmorphism in RASopathies, and so on.

The identification of LVH in a neonate or an infant with a matrilinear inheritance is highly suggestive of a metabolic or a mitochondrial disease [19]. On the other hand, the identification of LVH in an elderly patient represents a red flag for amyloidosis, particularly when associated with a discrepancy between the EKG voltages and the degree of LVH in the echocardiogram [24].

In addition to the clinical presentation and type of inheritance, EKG abnormalities might provide important diagnostic hints: for example, in patients with massive LVH in echocardiograms, high EKG voltages associated with pre-excitation are characteristic of Danon disease [19]. Other disease-specific EKG findings are short PQ intervals in Anderson–Fabry disease and atrio-ventricular blocks in cardiac amyloidosis or storage diseases [19].

In addition to the clinical observations and EKGs, conventional and advanced imaging plays a central role in the diagnostic workup of unexplained LVH cases and often shows early abnormalities and disease-specific signs, which may lead clinicians toward the correct diagnosis. This topic is discussed in more detail in paragraph 7.

## 4. Genetics of Sarc+ and Sarc− HCM

HCM has classically been recognized as a disease of the sarcomere (Figure 1C) [25]. Indeed, the most frequent genetic causes of HCM are mutations in genes encoding thick filament proteins, namely, myosin heavy chain-7 (MYH7) encoding for cardiac beta-myosin heavy chain (β-MHC), cardiac myosin binding protein C (MYBPC3), Myosin Light Chains 2 and 3 (MYL2 and MYL3), and myosin heavy chain 6 (MYH6) encoding cardiac alpha-myosin heavy chain (α-MHC), with the first two accounting for more than 70% of Sarc+ cases [26]. The genes encoding thin filament components, including cardiac α-actin 1 (ACTC1) and the troponin/tropomyosin complex formed by cardiac troponin C (TNNC1), cardiac troponin I (TNNI3), cardiac troponin T (TNNT2), and tropomyosin 1 (TPM1), are associated with less than 5% of Sarc+ HCM cases [16] (Figure 1C). Furthermore, mutations in genes coding for proteins with either structural or enzymatic functions located in other structures of the sarcomere, such as the Z disc proteins Alpha-actinin-2 (ACTN2) and Myozenin-2 (MYOZ2) [27] or the M line proteins Muscle RING Finger (MuRF1), Obscurin

(OBSCN) and Myomesin 2 (MYOM2), can occasionally be detected as genetic causes of Sarc+ HCM [26].

In some cases, the disease may be caused by the occurrence of more than one genetic variant: the presence of double heterozygous, compound heterozygous, and homozygous mutations is often associated with more severe disease [25]. After the introduction of next-generation sequencing (NGS), an increased number of genetic variants were detected in both sarcomere and non-sarcomere genes, allowing for the early identification of genetically affected family members and preventing the unnecessary follow-up of non-carriers. However, the NGS-based approaches have also increased the yield of variants of unknown significance (VUS), the clinical interpretation of which remains challenging [28].

According to the American College of Medical Genetics and Genomics (ACMG), a genetic variant can be considered pathogenic (P) or likely pathogenic (LP) if at least one of the following criteria is met [29]

- the genetic variant co-segregates with the HCM phenotype in the family and is absent in the phenotype-negative individuals;
- the genetic variant has prior evidence of pathogenicity, which means it has been documented as a disease-causing mutation in ≥1 patient in the published literature;
- the genetic variant is absent in the healthy population;
- the genetic variant is predicted (in silico or by functional studies) to cause major disruptions of the structure and function of the encoded protein.

In HCM, the causal mutation is often private (which means it is described only in one family) or is detected in small family pedigrees or may be a de novo variant identifiable only in the proband; moreover, HCM typically shows incomplete penetrance and variable expressivity of the phenotype, which may be due to the influence of environmental and genetic modifiers [29]. For these reasons, in many cases, the ACMG criteria cannot be used to establish the pathogenicity of genetic variants identified in HCM probands [29].

To identify the missing causal genes in Sarc− HCM, it is probably necessary to shift from a deterministic approach, which assumes that HCM is caused by a single mutation with a large effect, to a probabilistic approach, which considers HCM a polygenic disease in which multiple genetic variants with moderate effect sizes collectively contribute to the development of the phenotype (Figure 2A). Epigenetics, genetic variants with modifier effects, and responses to environmental factors are also expected to affect the expression of the phenotype (Figure 2B) [30].

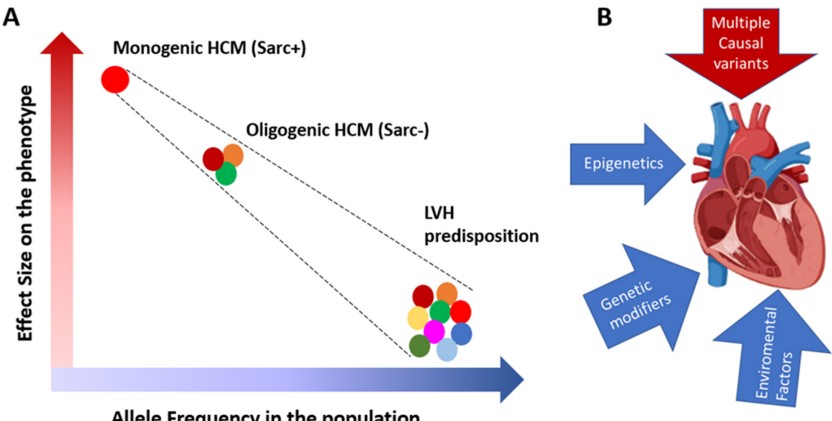

**Figure 2.** (**A**): Rare variants with large effect sizes are more common in familial monogenic HCM, while those with moder-ate effect sizes and higher population frequencies are often found in the sporadic cases and in small families with oligo-genic HCM. On the other side are the common genetic variants with small size effects which in combination may pre-dispose to LVH in the presence of an overload. (**B**): HCM phenotypic expression is the result of the interplay of the caus-al variants with epigenetics, genetic modifiers and environmental factors (for example obesity, and hypertension).

## 5. HCM Caused by Mutations in Non-Sarcomeric Genes

Advances in DNA-sequencing technologies and the introduction of NGS cardiomyopathy gene panels in clinical practice allow for the rapid analysis of many genes at affordable costs, providing the opportunity to identify the missing P/LP variants in many Sarc− HCM patients [30]. Therefore, although mutations in sarcomeric genes remain the most common causes of HCM, variants in several additional genes encoding non-sarcomeric proteins have been associated with the disease in a small number of HCM patients [31]. At the present time, the genes showing strong evidence of an association with HCM are: cysteine and glycine-rich protein 3 (CSRP3) [32,33], four and half LIM domains 1 (FHL1) [34,35], filamin C (FLNC) [36–38], formin homology 2 domain-containing 3 (FHOD3) [39,40], junctophilin 2 (JPH2) [41], phospholamban (PLN) [42,43], Tripartite Motif-Containing 63 (TRIM63) [44,45], and Kelch-like protein 24 (KLHL24) [46]. Table 1 shows the function of the proteins encoded by these genes and the level of evidence for their association with HCM.

**Table 1.** Non-sarcomeric genes associated with HCM, encoded protein's function and level of evidence for their as-sociation with HCM. AD: autosomal dominant; AR: autosomal recessive.

| Gene | Function of the Encoded Protein | Level of Evidence for HCM Association | Phenotypes and Mode of Inheritance | References |
|---|---|---|---|---|
| CSRP3 | regulation of myogenesis; maintenance of myocyte cytoskeleton; mechano-signaling and transduction | Moderate | HCM and rarely DCM (AD and AR) | [32,33] |
| FHL1 | Biomechanical sensing; regulation of sarcomere stiffness, hypertrophy, ion channels | Moderate | HCM and *FHL1*-related myopathies (X-linked) | [34,35] |
| FLNC | Crosslinking of actin filaments and interaction with Z-disc and sarcolemma | Limited | HCM, DCM and ACM (AD) | [36–38] |
| FHOD3 | Promoting polymerization of actin thin filaments | Limited | HCM and DCM (AD) | [39,40] |
| JPH2 | Coupling of transverse tubule associated L-type Ca2+ channels with RYR2 | Moderate | HCM (AD) and DCM (AR) | [41] |
| PLN | Regulation of sarco/endoplasmic reticulum Ca2+ ATPase activity | Limited | HCM, DCM and ACM (AD and AR) | [42,43] |
| TRIM63 | Regulation of sarcomeric protein degradation | Limited | HCM (AD and AR) | [44,45] |
| KLHL24 | Regulation of the balance between intermediate filament stability and degradation | Limited | Epidermolysis bullosa simplex with DCM (AD) HCM (AR) | [46] |

## 6. Patients' Demographic and Clinical Characteristics

Sarc+ HCM patients usually present with a younger age at diagnosis and a higher prevalence of family histories of HCM as compared with Sarc− patients [47]. The lower rate of clinically affected family members in the pedigrees of patients with Sarc− HCM suggests that, in this subpopulation of HCM patients, the disease may be caused by a germline mutation with a very low penetrance or, alternatively, be the consequence of the combined effect of multiple genetic variants [48]. In both Sarc+ and Sarc− HCM, the interaction of the causal mutations with genetic modifiers, epigenetic modifications, and environmental factors [49] may further influence the morphological and clinical phenotypical expression [47].

Sarc− patients present with a higher prevalence of hypertension and obesity as compared with Sarc+ patients [50]. This suggests a more noticeable effect of ageing, metabolic

dysfunction, and volume and/or pression overload on the development and degree of LVH in this HCM subpopulation [50].

It has been shown that obesity is an independent risk factor for LVH development in HCM, even in the presence of other comorbidities such as hypertension and diabetes [51–53]. Obesity has also been shown to be a predictor of long-term adverse events such as ventricular tachycardia or atrial fibrillation in HCM patients [54]. The association between obesity and LVH is corroborated by the evidence of the regression of LVH after aggressive weight loss [55]. Multiple mechanisms have been proposed regarding the role of obesity on LVH development, including hemodynamic factors (such as volume expansion and increased cardiac output), metabolic factors (such as insulin resistance and lipotoxicity) and the stimulation of cardiomyocyte growth by circulating hormones and cytokines released by dysfunctional fat [50,56].

The effects of ageing on LVH have been related to several mechanisms. Ageing-related increases in vascular stiffness increases the afterload, which stimulates LVH. In addition, ageing is associated with increased myocardial interstitial fibrosis and abnormal $Ca^{2+}$ handling; both of these age-related changes contribute to diastolic dysfunction [48].

Sarc− HCM patients are more often males, older at the onset of atrial fibrillation, and show reduced incidence of non-sustained ventricular tachycardia and a lower risk of SCD as assessed by the European Society of Cardiology (ESC) HCM Risk SCD score, as compared with the Sarc+ HCM patients. Because of their lower SCD risk score, the Sarc− patients are also less likely to receive an implantable cardioverter defibrillator (ICD) for primary prevention as compared with Sarc+ patients [47]. The demographic, clinical, and imaging differences between Sarc+ and Sarc− HCM are summarized in Table 2.

**Table 2.** Differences characteristics, echocardiography and Cardiac MRI findings in Sarc+ and Sarc− HCM patients [15].

| Patient Characteristics | Sarc+ | Sarc− |
|---|---|---|
| **Male sex** | +− | ++ |
| **Age at diagnosis** | 46 ± 12 | 51 ± 10 |
| **Hypertension** | +− | ++ |
| **Obesity** | +− | ++ |
| **Comorbidities** | + | ++ |
| **Susteined VT** | ++ | +− |
| **Sudden death event** | ++ | − |
| **High HCM−risk score** | ++ | − |
| **ICD** | ++ | − |
| **Atrial fibrillation** | +− | +− |
| **Echocardiography** | | |
| **Maximal wall thickness** | ++ | +− |
| **Isolated basal septal hypertrophy** | +− | ++ |
| **Apical hypertrophy** | +− | ++ |
| **Reverse septal morphology** | ++ | − |
| **LVOTO** | +− | ++ |
| **Dilated LA** | +− | +− |
| **Cardiac MRI** | | |
| **Presence of LGE** | ++ | −+ |
| **ECV expansion** | ++ | −+ |

The Mayo Clinic HCM Genotype Predictor score (Mayo Score), published in 2014, has been proposed for predicting the diagnostic yield of a positive genetic test on the bases of clinical and echocardiographic variables. The Mayo Score algorithm is based on 5 positive predictors and 1 negative predictor of positive genetic test results: age at diagnosis younger than 45 years, family history of HCM and/or SCD, a reverse-curve morphology of the septum, and a maximal wall thickness (MWT) ≥ 20 mm are considered positive predictors, whereas the presence of hypertension is considered a negative predictor [57]. Since the prediction power of the Mayo Score ranges from 6%, when only hypertension is present, to 80%, when all five positive predictors are present [57], an effective prediction of genetic tests results could be achieved using this phenotype-based score, although the final diagnosis may be reached only after the genetic screening.

## 7. Echocardiography and Cardiac Magnetic Resonance Findings

Echocardiography and cardiac magnetic resonance (CMR) are first-line diagnostic techniques for HCM and may help to differentiate Sarc− patients from Sarc− patients (Table 2). These two HCM subpopulations may show different LVH distributions: isolated basal septal hypertrophy or a concentric pattern should raise suspicions of a Sarc− form, while the reverse curvature of the septum is more common in the Sarc+ group [47]. Indeed, it has been shown that, independently of age, the morphology of the septum strongly predicts the presence or absence of HCM-associated sarcomeric mutations [58]. The different morphologies of the septum could explain the higher proportion of patients with a significant LVOT gradient in the Sarc− subgroup as compared to the Sarc+ group [15]. This is a surprising observation, since LVOT obstruction (LVOTO) was considered one of the most frequent echocardiographic findings in classic Sarc+ HCM [59].

Other echocardiographic differences between Sarc+ and Sarc− HCM are a greater MWT and smaller LV end-diastolic (LVEDD) and end-systolic diameters (LVESD) in Sarc+ [47] and a higher prevalence of apical hypertrophy (ApHCM) in Sarc− HCM [15]. First described in Japan in 1976 and not as rare as first thought (accounting for up to 25% of HCM in Asian populations and up to 10% in non-Asian HCM) [60], ApHCM is characterized by "giant" negative precordial T-waves on electrocardiograms and by the "spade-like" configuration of the LV cavity in the end-diastole in 2D echocardiography [61]. This morphology was originally thought to be associated with an increased mortality risk, but recent data suggest annual cardiac death rates of 0.5% to 4%, similar to those seen in classic HCM [62,63].

More sensitive imaging techniques, such as speckle tracking and strain analysis by echocardiography and cardiac magnetic resonance (CMR) with tissue characterization by late gadolinium enhancement (LGE) and T1 and T2 mapping, might enable the identification of subclinical cardiac involvement (such as interstitial fibrosis and disarray) in the early stages, when a clear LVH is not yet evident. Myocardial fibrosis is one of the histopathologic hallmarks of HCM and its presence has been associated with an increased risk of SCD, ventricular tachyarrhythmias, LV dysfunction, and heart failure [63–67]. However, the pathogenesis of interstitial fibrosis in HCM remains unknown. Myocardial fibrosis may be evaluated non-invasively by CMR after injection of an intra-venous bolus of gadolinium. This contrast agent accumulates in the extracellular matrix of both normal and abnormal myocardia; however, in the later acquisitions, because of slower kinetics and a larger volume of distribution, the abnormal myocardium shows larger amounts per unit of volume of gadolinium, the so-called LGE. In Sarc+ individuals, the LGE is more frequently detected and, when present, is often more severe than in Sarc− HCM patients [15] (Table 2).

Another CMR finding typical of HCM is the evidence of abnormal extracellular volume (ECV). Cardiac MRI with pre- and postcontrast T1 mapping demonstrates a significant increase in myocardial ECV in Sarc+ carriers, also before LVH development [68]. This observation suggests that HCM sarcomere mutations lead to myocardial abnormalities, potentially triggering interstitial fibrosis, independently of the presence and degree of LVH [68].

By combining information regarding demographics, family history, the presence of comorbidities and complications, the degree and distribution of LVH, and the amount and localization of LGE, the clinician may be steered toward a diagnosis of Sarc+ or Sarc− HCM (Figure 3). Of course, a final and definitive etiological definition can be reached only after examining the genetic test results.

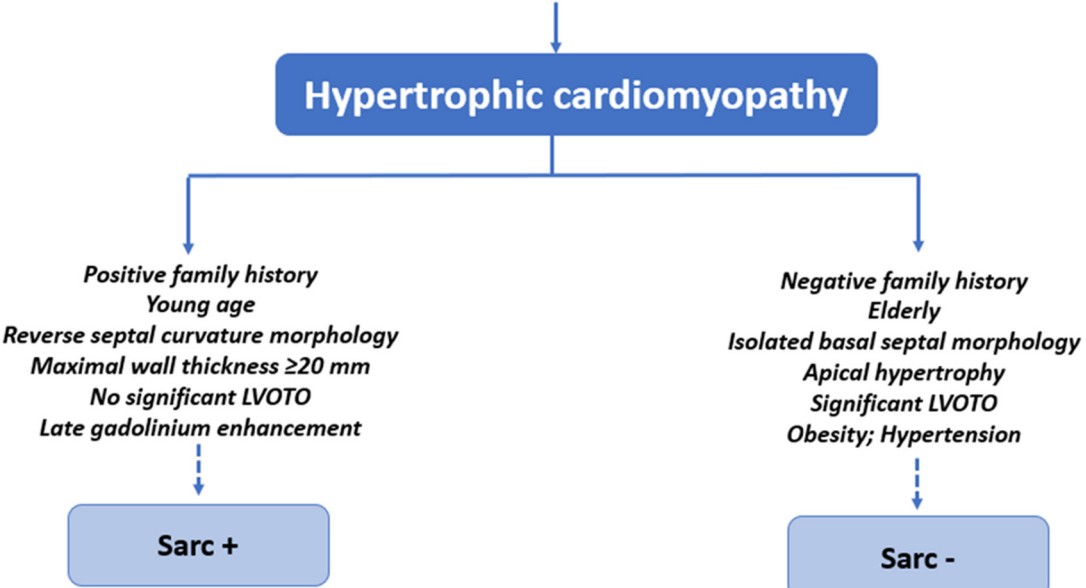

**Figure 3.** Demographic information, family history, presence of comorbidities and compli-cations, echocardiographic and CMR findings could orientate toward Sarc+ or Sarc− HCM.

## 8. Clinical Presentation and Prognosis

HCM patients can be categorized into different prognostic profiles, with only 10% of HCM patients experiencing one or more of adverse pathways [69]. These profiles should not be considered to represent degrees of severity of the disease, but rather a patient- or family-specific clinical outcomes.

The main clinical profiles observed in HCM patients are: (1) patients with a stable and benign clinical profile, who do not require treatment [70]; (2) patients with LVOTO and significant heart failure (HF) symptoms, who need medical or invasive interventions to reduce the gradient and HF symptoms [71]; (3) patients with increased SCD risk, who are candidates for an ICD implant in primary prevention [72]; (4) patients with atrial fi-brillation and increased risk of embolic stroke and a strong indication for anticoagula-tion [73]; and (5) patients with nonobstructive end-stage dilatative evolution with reduced systolic function, who need advanced HF therapies [74].

Survival studies have demonstrated that Sarc+ patients experience the significantly earlier onset of events, a higher incidence of HF and atrial fibrillation, higher risk of ven-tricular arrhythmias, and worse outcomes, with at least twice the lifetime hazard of death as compared with Sarc− patients [75]. It has been shown that, in Sarc− patients with no family history of HCM, the overall mortality for all causes is similar to that of the general population [47].

According to the 2020 American College of Cardiology/American Heart Association (ACC/AHA) and the 2014 ESC guidelines on HCM clinical management, the main role of genetic testing in these patients should be to drive the cascade screening of the family members [2,14], rather than to guide the clinical management and the prognostic stratification of the probands [2]. However, the 2022 ESC guidelines for the management of pa-tients with ventricular arrhythmias and the prevention of sudden cardiac death consider the presence of a sarcomeric mutation as an important factor affecting the decision to implant an ICD in HCM, mainly in the patients who are at intermediate risk of SCD. Indeed, according to the latest ESC guidelines, an ICD implantation is indicated in the presence of a 5-year HCM Risk-SCD score $\geq$6% (high risk), while an ICD should be considered in HCM patients aged 16 years or more with a 5-year HCM Risk-SCD score between 4 and 5 (intermediate risk) and at least one of the following characteristics:

- significant LGE at CMR (usually $\geq$15% of LV mass);
- LVEF < 50%;
- abnormal blood pressure response during exercise test;
- LV apical aneurysm;
- presence of a sarcomeric pathogenic variant [76].

Hence, the latest guidelines and several published studies suggest a better prognosis in Sarc− HCM as compared with Sarc+ HCM. In addition to patients' lower personal risk of experiencing adverse events, Sarc− HCM has also been associated with a low probability of clinically relevant phenotypic manifestations in family member [77]. In fact, it has been shown that, if an HCM proband is Sarc−, the risk of the disease developing in his or her first-degree relatives is less than 50%; for this reason, in this subpopulation of HCM, the long-term clinical surveillance of adult relatives is not necessary, while a re-evaluation of the results of genetic test must be undertaken periodically for the possible reclassification of a VUS into a pathogenic variant [77].

Based on these observations, the clinical screening of Sarc− HCM could be less strict than that of the Sarc+ HCM subpopulation, and the focus should rather be the manage-ment of comorbidities such as overweight and hypertension [52].

## 9. Phenotypic Variability and Personalized Clinical Approach

A notable feature of HCM is its considerable phenotypic variability, including both morphologic characteristics, such as the LVH degree and distribution, and clinical manifestations.

The determinants of this phenotypic variability are largely unknown. The causal mutation, while essential for the expression of the phenotype, does not fully explain the variability in hypertrophic expressivity. This is best illustrated in familial HCM, wherein the affected individuals who share a common causal mutation exhibit variable degrees of cardiac hypertrophy.

A genome-wide mapping study, including 811 short tandem repeat markers of an HCM family of 100 members including 36 carriers of the InsG791 mutation in MYBPC3 identified 4 modifier chromosomal loci, responsible for the variation in hypertrophic expressivity [8]. The findings from this study indicate that genes other than the causal gene may contribute to the variable expression of the cardiac phenotype in HCM [8], suggesting that the genetic background, including all the modifier genes, might have significant effects on the phenotypic manifestations induced by the causal mutation. However, in this study, only modifier chromosomal loci, each containing many genes, were identified. Several polymorphism association studies, which explore the association of functional single nucleotide polymorphisms (SNPs) in candidate genes, have been performed to identify the individual genetic modifiers. However, the results of these studies are inconsistent because of the confounders frequently encountered in SNP association studies, such as small sample sizes, biological plausibility, population characteristics, association strength, and so on. Recently, genome-wide studies and multi-omics studies have been performed on large HCM cohorts through the international collaboration of several centers, and the

genetic modifiers and cellular and molecular mechanisms of HCM have started to be unraveled [10,11,78].

Overall, the genomics studies suggest that the final phenotype in HCM is the result of the interactions between the causal genes, the genetic background, comorbidities, and environmental factors. Thus, the identification and characterization of patient-specific genetic profiles, together with a personalized comprehensive clinical evaluation, are expected to constitute the approach to the management of HCM patients and the foundation for the development of more effective therapies in the future.

## 10. Conclusions

Distinct disease subsets are currently clustered under the same broad designation of HCM. However, several studies have shown significative differences between the Sarc+ and Sarc− HCM subpopulations in terms of demographic, clinical (including prognosis), and imaging characteristics. It is plausible that the differences observed between these two groups are determined by their different genetic backgrounds. In Sarc+ HCM, the main mechanism for the development of the disease is the dominant negative effect of the mutant protein that is incorporated into the sarcomere and causes sarcomeric dysfunction in terms of relaxation and ATPase activity. These initial defects activate signaling pathways that induce the typical histological changes observed in HCM, such as myocyte disarray and hypertrophy, interstitial fibrosis, and small vessel disease. These molecular and histological changes ultimately lead to the clinical manifestations of HCM, such as asymmetrical cardiac hypertrophy, diastolic dysfunction, arrhythmias, and heart failure.

On the other hand, Sarc− HCM includes a more heterogeneous group of patients who are carriers of mutations in different non-sarcomeric genes and sometimes of a combination of mutations in more than one causal gene. For this reason, the pathogenic mechanisms of Sarc− HCM are highly diverse, depending on the function of the causal genes. As the published studies' outcomes are based on the overall Sarc− HCM subpopulation rather than on individual cases, we can hypothesize that less severe outcomes are reported in this group in the literature because, in most cases, the causal genetic variants might have modest effects on the molecular pathway and, consequently, on the development of histological and clinical phenotypes. However, even though the overall Sarc− population is considered to be at lower risk than the Sarc+ population, some rare non-sarcomeric forms of HCM are characterized by a high incidence of SCD and evolution toward HF.

The establishment of international registries is necessary to better characterize rare forms of HCM and improve the risk stratification and clinical management of HCM patients through more personalized preventive and therapeutic approaches.

**Author Contributions:** Conceptualization: F.B., R.L., G.C. and M.A.L.; methodology: F.B., G.C., G.T. and E.F.P.; validation: R.L., G.E. and M.A.L.; software: G.T. and L.O.; Investigation G.C., R.L., G.F., G.E. and M.A.L. All authors have read and agreed to the published version of the manuscript.

**Funding:** This research received no external funding.

**Institutional Review Board Statement:** Not applicable.

**Informed Consent Statement:** Not applicable.

**Data Availability Statement:** No new data were created.

**Conflicts of Interest:** The authors declare no conflict of interest.

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
