# Peer review of "Sarcomeric versus Non-Sarcomeric HCM"

_cardiogenetics, doi:10.3390/cardiogenetics13020009_

Round 1

Reviewer 1 Report

Overall, this is a comprehensive review regarding sarcomeric and non-sarcomeric cardiomyopathies. To my knowledge there has been no such comparison of these diseases, so the topic is well covered and above all interesting.  I find it to be well structured and appropriately divided into specific sections. Few points that need to be addressed are:

1.       Line 83: Abbreviations for sarcomeric (Sarc+) and non-sarcomeric (Sarc-) were previously introduced so it is redundant to repeat complete names

2.       Lines 80-100: missing references for an entire paragraph

3.       Line 167: repetition of line 159

I recommend next changes:

1.       Line 271: would  instead of will

2.       Line 276: word ONE instead of 1

3.       To consider citing Melas et al. https://doi.org/10.3390/jcm12010225 in sections 4 or 5

However, the main drawback was the style/English language. Some sentences were difficult to understand, because it looked like expressions from the mother tongue were just translated into English. The manuscript is rich in colloquial expressions, the use of which is frivolous for a scientific journal. Examples are:

1.       Line 119: ”but not always more means better”

2.    Line 129: “Absence from large, ethnicity-matched healthy population;” - sincerely, I don’t understand this sentence

3.       Line 172: “after controlling for…”maybe better expression would be checking

4.       Line 185:  “major responsible” the most responsible?

5.       Line 317: Whole sentence is example of poor English

6.       Lines 122, 132, 223..: More words in quotation marks than necessary

General comment: The manuscript needs general revision for language, style and grammar

Author Response

Overall, this is a comprehensive review regarding sarcomeric and non-sarcomeric cardiomyopathies. To my knowledge there has been no such comparison of these diseases, so the topic is well covered and above all interesting.  I find it to be well structured and appropriately divided into specific sections.

Response: We thank the reviewer for the positive comments on the review and for her/his inputs which have significantly improved the manuscript.

Few points that need to be addressed are.

  1. Line 83: Abbreviations for sarcomeric (Sarc+) and non-sarcomeric (Sarc-) were previously introduced so it is redundant to repeat complete names.

Response: We have checked the whole manuscript after the first introduction of sarcomeric (Sarc+) and non sarcomeric (Sarc-) and we found several redundant complete names for these which we have corrected to the abbreviated forms Sarc+ and Sarc- as suggested by the reviewer.

  1. Lines 80-100: missing references for an entire paragraph.

Response: We have added the missing references. Due to the expensive English editing and the addition of more paragraphs as suggested by the other reviewers, the previous paragraph including lines from 80 to 100 is now from line 105 to line 131.

  1. Line 167: repetition of line 159

Response: We have eliminated the repetition. Due to the expensive English editing and the addition of more paragraphs as suggested by the other reviewers, line167 is now Line 216, while line 159 is now line 207.

I recommend next changes:

  1. Line 271: would instead of will

Response: We have corrected as suggested by the reviewer. Due to the expensive English editing and the addition of more paragraphs as suggested by the other reviewers, line 271 is now line 328.

  1. Line 276: word ONE instead of 1 FATTO

Response: we changed the number 1 with the word one (line 276 is now line 334).

  1. To consider citing Melas et al. https://doi.org/10.3390/jcm12010225in sections 4 or 5

Response: We agree with the reviewer and cited this is interesting paper in paragraph 4, at lines 134 and156.

However, the main drawback was the style/English language. Some sentences were difficult to understand, because it looked like expressions from the mother tongue were just translated into English. The manuscript is rich in colloquial expressions, the use of which is frivolous for a scientific journal. Examples are:

  1. Line 119:” but not always more means better”
  2.   Line 129: “Absence from large, ethnicity-matched healthy population;” - sincerely, I don’t understand this sentence FATTO
  3. Line 172: “after controlling for…” maybe better expression would be checking
  4. Line 185: “major responsible” the most responsible?
  5. Line 317: Whole sentence is example of poor English
  6. Lines 122, 132, 223.: More words in quotation marks than necessary

Response: A comprehensive revision of the English language/style has been done and all the suggestions and clarifications requested by the reviewer have been taken care of.

General comment: The manuscript needs general revision for language, style, and grammar.

Response: A comprehensive revision of the English language/style and grammar has been done.

Reviewer 2 Report

The review presented by Borrelli F et al on sarcomeric versus non-sarcomeric HCM adds one more review in the field and is a understandable paper. One interesting point is the comparison between sarcomeric and non-sarcomeric gene mutations or variants, earlier et late onset of HCM as well as their related phenotype difference. Figure 2 is a good idea. The weak point is lacking more concise synthesized overview on the different data and potential explanation. 

The references of Table 1 are wrong in number. Some abbreviations appear before the complete definition such as CMR.

Author Response

The review presented by Borrelli F et al on sarcomeric versus non-sarcomeric HCM adds one more review in the field and is a understandable paper.

One interesting point is the comparison between sarcomeric and non-sarcomeric gene mutations or variants, earlier et late onset of HCM as well as their related phenotype difference. Figure 2 is a good idea.

Response: We thank the reviewer for the positive comments and for her/his suggestions which have improved our manuscript.

The weak point is lacking more concise synthesized overview on the different data and potential explanation.

Response: We agree with the reviewer; hence, we have modified the last paragraph “Conclusions” underlying the main differences between Sac+ and Sarc- HCM and hypothesizing the possible mechanisms leading to these differences. 

The references of Table 1 are wrong in number. Some abbreviations appear before the complete definition such as CMR.

Response: We have corrected the references of Table 1. Moreover, we have checked that the complete definitions are present before the abbreviations when they appear for the first time in the text.  

Reviewer 3 Report

The manuscript, "Sarcomeric versus non-sarcomeric HCM” represents a concise review article describing the differences in phenotypic expression and clinical outcomes of Sarc+ and Sarc- HCM and our current knowledge about HCM caused by rare non-sarcomeric mutations.

The entire manuscript is well written in all its parts, supplemented by a well-chosen illustrative and tabular data presentation, and accompanied by an appropriate and up-to-date reference list.

In my opinion, the presented review will be of great interest to readers of Cardiogenetic from both clinical and basic biomedical science fields.

I congratulate the authors for their effort in writing this concise, well-written, easy-to-follow, and highly informative review.

Author Response

The manuscript, "Sarcomeric versus non-sarcomeric HCM” represents a concise review article describing the differences in phenotypic expression and clinical outcomes of Sarc+ and Sarc- HCM and our current knowledge about HCM caused by rare non-sarcomeric mutations.

The entire manuscript is well written in all its parts, supplemented by a well-chosen illustrative and tabular data presentation, and accompanied by an appropriate and up-to-date reference list.

In my opinion, the presented review will be of great interest to readers of Cardiogenetic from both clinical and basic biomedical science fields.

I congratulate the authors for their effort in writing this concise, well-written, easy-to-follow, and highly informative review.

Response: We thank the reviewer for his positive comments on the review.